# Disruption of dopamine D2/D3 system function impairs the human ability to understand the mental states of other people

Bianca A. Schuster[1,2]*, Sophie Sowden[1], Alicia J. Rybicki[1], Dagmar S. Fraser[1], Clare Press[3,4], Lydia Hickman[1,5], Peter Holland[6], Jennifer L. Cook[1]

1 Centre for Human Brain Health and School of Psychology, University of Birmingham, Birmingham, United Kingdom, 2 Department of Cognition, Emotion, and Methods in Psychology, University of Vienna, Vienna, Austria, 3 Department of Psychological Sciences, Birkbeck University of London, London, United Kingdom, 4 Wellcome Centre for Human Neuroimaging, UCL, London, United Kingdom, 5 MRC Cognition and Brain Sciences Unit, University of Cambridge, Cambridge, United Kingdom, 6 Department of Psychology, Goldsmiths University of London, London, United Kingdom

* biancaschuster05@gmail.com

**Data Availability Statement:** All data and code files are publicly avaiable on OSF via the link https://osf.io/xm7ty/ (DOI 10.17605/OSF.IO/XM7TY).

## Abstract

Difficulties in reasoning about others' mental states (i.e., mentalising/Theory of Mind) are highly prevalent among disorders featuring dopamine dysfunctions (e.g., Parkinson's disease) and significantly affect individuals' quality of life. However, due to multiple confounding factors inherent to existing patient studies, currently little is known about whether these sociocognitive symptoms originate from aberrant dopamine signalling or from psychosocial changes unrelated to dopamine. The present study, therefore, investigated the role of dopamine in modulating mentalising in a sample of healthy volunteers. We used a double-blind, placebo-controlled procedure to test the effect of the D2/D3 antagonist haloperidol on mental state attribution, using an adaptation of the Heider and Simmel (1944) animations task. On 2 separate days, once after receiving 2.5 mg haloperidol and once after receiving placebo, 33 healthy adult participants viewed and labelled short videos of 2 triangles depicting mental state (involving mentalistic interaction wherein 1 triangle intends to cause or act upon a particular mental state in the other, e.g., surprising) and non-mental state (involving reciprocal interaction without the intention to cause/act upon the other triangle's mental state, e.g., following) interactions. Using Bayesian mixed effects models, we observed that haloperidol decreased accuracy in labelling both mental and non-mental state animations. Our secondary analyses suggest that dopamine modulates inference from mental and non-mental state animations via independent mechanisms, pointing towards 2 putative pathways underlying the dopaminergic modulation of *mental* state attribution: action representation and a shared mechanism supporting mentalising and emotion recognition. We conclude that dopaminergic pathways impact Theory of Mind, at least indirectly. Our results have implications for the neurochemical basis of sociocognitive difficulties in patients with dopamine dysfunctions and generate new hypotheses about the specific dopamine-mediated mechanisms underlying social cognition.

**Funding:** This work was supported by the European Union's Horizon 2020 Research and Innovation Program (https://erc.europa.eu/faq-programme/horizon-europe-horizon) under ERC-2017-STG Grant Agreement No. 757583 (Brain2Bee, awarded to J.L.C.) The funders did not play any role in the study design, data collection and analysis, decision to publish, or preparation of the manuscript.

**Competing interests:** The authors have declared that no competing interests exist.

**Abbreviations:** CrI, credible interval; HD, Huntington's disease; PD, Parkinson's disease; PFC, prefrontal cortex; PLW, point light walker; PwP, people with Parkinson's; ToM, Theory of Mind; TS, Tourette's syndrome; WM, working memory.

## Introduction

Sociocognitive difficulties are common among disorders featuring dopamine dysfunction, such as Parkinson's disease (PD) [1], Huntington's disease (HD) [2], Tourette's syndrome (TS) [3], and schizophrenia [4]. Such difficulties typically include challenges with attributing and understanding mental states (i.e., putting oneself in others' shoes, also referred to as mentalising or Theory of Mind [ToM] [5]). Alarmingly, mentalising difficulties in the aforementioned populations are consistently associated with negative outcomes including increased disease burden and poor quality of life [6–8], but little is understood about their aetiology. Disorders that feature dopamine dysfunction are commonly linked with wider psychosocial changes including social isolation and withdrawal, and mentalising difficulties may plausibly stem from these psychosocial changes. However, a powerful, underexplored alternative is that dopamine is *causally implicated* in mentalising. The current literature lacks direct empirical evidence for a causal role of dopamine in ToM. To this end, we investigate the effect of a dopamine-modulating drug on the mentalising performance of healthy members of the general population and further explore several potential mechanistic pathways.

Inconsistencies in the existing literature mean that there is currently no empirical consensus supporting a causal role for dopamine in mentalising. For instance, to our knowledge, there is only 1 study directly comparing ToM abilities within people with Parkinson's (PwP) on, and after acute withdrawal of, dopaminergic medication [9]. This study found no differences in ToM performance between drug on and off states, and performance was comparable to healthy controls in a subsample of early-stage PwP. These data, therefore, do not causally implicate dopamine in mentalising. However, the relatively preserved ToM function of PwP in the drug off state (i.e., putatively low dopamine) may also indicate an insufficient washout period or cognitive and/or neural compensation strategies in the early PD sample [10]. In support of this, a more recent study showed ToM differences in early stage drug-naïve PwP compared to control participants, which improved after 3 months of dopaminergic therapy [11]. Evidence from studies examining those with schizophrenia is equally inconclusive: For instance, 1 study [12] reported improvements in ToM ability in patients treated with certain atypical antipsychotics (e.g., olanzapine), and detrimental effects of typical antipsychotic medication (e.g., haloperidol), but putative selection bias and the lack of on-off comparisons make it impossible to clearly attribute group differences to effects of the dopaminergic drugs. Other inconsistencies, including methodological differences, as well as high between- and within-study variance in disease stage, medication, and comorbidities [11], make it difficult to draw clear inferences solely from patient studies.

An incisive way to establish a causal role is to observe the influence of dopaminergic drugs on mentalising in the healthy population. However, psychopharmacological data are scarce. While many studies have shown effects of dopamine disruption on general cognition (also referred to as "neurocognition," e.g., attention, learning, and executive function [13]), the literature is less clear regarding influences on *social* cognition. While a handful of existing studies show effects of dopamine D2/D3 receptor antagonism on emotion recognition [14,15] and social belief updates [16–19], to the best of our knowledge, no published study to date has explored effects of dopamine manipulation on ToM function in healthy individuals.

There are several candidate pathways that could underpin a causal role of dopamine in social cognition. Dopamine is implicated in cognitive control functions, including working memory, attention, and flexible behaviour, via its neuromodulatory actions on the prefrontal cortex (PFC) [20]. Thus, any effect of dopaminergic manipulation on mentalising could, at least in part, arise from a decreased ability to maintain and manipulate mental state representations. Yet, this is perhaps unlikely to constitute the full mechanistic explanation since studies

often show that ToM deficits are independent from (mild) cognitive dysfunction [7,21–24]. A second (nonmutually exclusive) hypothesis relates to action simulation. A growing body of work suggests that brain areas involved in the planning and execution of actions also respond to the observation of others' actions, with the strength of the response being modulated by the observer's familiarity with the action [25–27]. In other words, observing others' actions automatically activates sensory-motor representations of one's own movements (motor codes required to produce the same action as well as anticipatory visual codes of the upcoming action in the sequence) in the observer. It has been suggested that humans use the same forward models for predicting the consequences of one's own movements to estimate the internal states (e.g., intentions, mental states) underlying others' movements [28], and research indicates that higher overlap between the low-level features of one's own and the observed action promotes higher accuracy in identifying those states [29–32]. Indeed, our previous work illustrated that similarity in movement between observer and actor facilitated mental state attribution in a classical mentalising task [30]. Recent experimental evidence has accumulated to support a role for dopamine in movement—including a role in movement vigour [33,34] as well as the control of movement kinematics [35–37]—that may be independent of its function in learning. Thus, individuals with dopamine dysfunction may exhibit differences in planning, preparation, and/or execution of movements, and these differences may contribute to difficulties in interpreting the actions of others with movements unaffected by dopamine disturbances.

Thus, using our own previously developed [30] version of a well-established mentalising task—the animations task, wherein participants label videos depicting mental (e.g., seducing) and non-mental (e.g., following) states—here, we first employed a pharmacological dopamine manipulation in healthy volunteers to investigate whether disruption of dopamine system function plays a causal role in mentalising. Second, by indexing effects of dopamine challenge on executive function (specifically working memory), motor function, and emotion recognition, we elucidate potential mechanistic pathways via which dopamine may modulate mentalising ability.

## Materials and methods

### Participants

Forty-three healthy volunteers (19 females; mean (M) [SD] age = 26.36 [6.3]) took part on at least one of 2 study days after passing an initial health screening. Participants were recruited via convenience sampling from University of Birmingham campus and city centres, gave written informed consent, and received either money (£10 per hour) for their participation. Five participants (2 placebo, 3 haloperidol) dropped out of the study after completing the first day, a further 5 could not complete the second test day due to COVID-19-related University closures, and consequently, all analyses are based on 33 full datasets. All experimental procedures were approved by the University of Birmingham Research Ethics Committee (ERN 18–1588) and Clinical Research Compliance Team and performed in accordance with the WMA Declaration of Helsinki (1975).

### Pharmacological manipulation and general procedure

Participants' eligibility for the study was evaluated by a clinician via review of their medical history, electrocardiogram assessment, and blood pressure check (see S1 Text for full details of inclusion/exclusion criteria). The main study took place on 2 separate test days, 1 to 4 weeks apart, where participants first completed an initial blood pressure and blood oxygenation check with the medic. Subsequently, in a double-blind, placebo-controlled within-subjects design, each participant took part on 2 study days, wherein all participants received tablets

containing either 2.5 mg haloperidol or lactose (placebo) on the first day, and the respective other treatment on the second day (order of drug day counterbalanced). For this, an independent researcher from the team pre-prepared an envelope with either placebo or haloperidol tablets. Participants were informed that none of the experimenters in the study knew the contents of the envelope. The independent researcher placed the capsules in the participants' hand and asked them to close their eyes before swallowing the tablets. Haloperidol is a dopamine D2/D3 receptor antagonist, which affects dopamine transmission via binding either to postsynaptic D2 and D3 receptors (blocking the effects of phasic dopamine bursts) and/or to presynaptic autoreceptors (which has downstream effects on the release and reuptake of dopamine and thus modulates bursting itself [38,39]).

Reported mean values for peak concentration and elimination half-life of oral haloperidol lie between 1.7 and 6.1 and 14.5 and 36.7 hours, respectively [40]. After drug or placebo administration, participants rested for 1.5 hours to allow for drug metabolization. Participant reported adverse responses were rare (5 out of 43 participants) and generally mild, with the most frequent symptoms mentioned being fatigue and headache. Importantly, only 3 of the 5 participants reported side effects after having received haloperidol.

Subsequently, participants began the task battery, which included the animations task, an emotion recognition task, a visual working memory task, and a movement task (see Tasks and procedure). Throughout the day, participants' blood pressure and oxygenation and arousal levels were checked hourly between tasks. All data were collected at the Centre for Human Brain Health (CHBH) at the University of Birmingham, United Kingdom.

## Tasks and procedure

Participants completed a task battery including tasks not described in this study (e.g., [14,19]). All relevant tasks are outlined below in the order they were presented to participants. Task order was the same on both study days.

## Visual working memory (WM) task

Participants completed an adaptation of the Sternberg [41] visual WM task, requiring them to determine whether a presented target letter was part of a previously displayed string of letters (varying in length from 5 to 9 consonants). This task is described in more detail in our previous study [14].

## Animations task

To assess drug effects on mentalising ability, we used a classical task that has been widely used in the literature for its sensitivity in detecting differences in mentalising performance between control and clinical groups where other tasks have failed to do so [42]: Animations tasks typically involve participants viewing and interpreting short videos of interacting triangles, which have been animated to either display so-called mental state interactions (i.e., mentalistic interaction wherein 1 triangle intends to cause or act upon a particular mental state in the other, e.g., "surprising"; in prior research referred to as "ToM" interactions) or non-mental state interactions (also called "goal-directed" interactions; involving reciprocal interaction without the intention to cause/act upon the other triangle's mental state, such as "following"). Within each mental state category, 2 words describing the interactions were chosen in equivalence to the words used in a seminal study by Abell and colleagues [42] and multiple following studies [43–46]: mental state: seducing, surprising; non-mental state: following, fighting. The key distinction between the 2 conditions is that the mental state (ToM) animations entail propositional attitudes wherein 1 agent intends to cause or act upon a particular mental state in the

other, while non-mental state (goal-directed) animations do not require such causal inference [47]. This is corroborated by prior research where the latter have been shown to consistently elicit lower levels of spontaneous attributions of intentionality than the mental state animations [43–45]. To evaluate motor contributions to mental state inference, we additionally asked participants to produce their own animations. Task setup and procedure were largely the same as in our previous study [30]: Participants both created and viewed animations on a WACOM Cintiq 22 HD touchscreen, tilted at an angle of approximately 30 degrees on a desk. They first created their own set of 35-second animations (2 *mental state*: seducing, surprising; 2 *non-mental state*: following, fighting) by moving 2 triangles on the touchscreen using their 2 index fingers, while positional data of both triangles were recorded at a frame rate of 133 frames/second. Subsequently, participants viewed and rated seducing, surprising, following, and fighting animations that had been created by an independent sample of participants (animation stimuli were the same as in our previous study [30]). Following each animation, participants rated on 4 separate visual-analogue scales (ranging from 1 [not at all] to 10 [very]) how much they thought the last viewed stimulus depicted the target, as well as each non-target word. For each of the target words, participants viewed 8 animations, resulting in a total of 32 animations, presented in pseudorandom order, on each study day. As in our previous study [30], the 8 animations were selected to represent the full speed frequency distribution of the stimulus pool, thus reflecting the full range of population kinematics. Note that, due to the pseudorandom selection of animation stimuli on each study day, animations viewed by each participant in haloperidol trials were not necessarily the same as in placebo trials.

As in our previous study [30], accuracy for each trial was calculated by subtracting the mean rating for all non-target words from the rating for the target word. Thus, a positive score indicates that the target word (e.g., surprising) was rated higher than the average of all non-target words (e.g., seducing, following, fighting) with higher positive accuracy scores reflecting better discrimination between target and non-target words and lower or negative accuracy scores representing high confusion between scales.

### Dynamic whole-body emotion recognition task

Participants viewed a total of 48 whole-body point light displays of male and female actors modelling angry, happy, and sad emotional walks (point light walkers [PLWs]; adopted from Edey and colleagues' study [31]). Following each stimulus, participants rated on 3 separate visual-analogue scales (ranging from 1 [not at all] to 10 [very]) how intensely they felt the stimulus expressed an angry, happy, or sad emotion. In line with the literature demonstrating that sadness is conveyed via slow, sluggish movements, anger with fast, jerky kinematics, and happiness intermediate to the two [48–51], sad PLWs exhibited the slowest mean speed, followed by happy, and then angry PLWs [52]. The task is described in more detail in our previous study [14].

### Movement task

Participants were asked to walk continuously between 2 sets of cones (placed 10 metres apart) for 120 seconds at their preferred walking speed. Acceleration data were recorded, using an iPhone 6s attached to the outer side of each participant's left ankle, with the app SensorLog [53]. To obtain an estimation of mean walking speed across the whole walk, first individual mean speed per pass was calculated by dividing the pass length (10 m) by the time taken to walk from one set of cones to another. Following this, speed estimates for all passes were averaged across the whole walk.

## Statistical analyses

All data were processed in MATLAB R2022a [54] and analysed with Bayesian mixed effects models using the brms [55] package in R [56]. Prior to model building, any continuous predictors were normalised and centred to allow comparisons between individual estimates. In what follows, we report our findings in terms of Bayesian credible intervals (CrIs, the Bayesian analogue of the classical confidence interval, with the exception that probability statements can be made based on CrIs [57]) and the posterior probability of models with and without an effect of interest (e.g., main effects or interactions). In brief, we used a standard analysis package (brms [55]) to assess the evidence for alternative models using a leave one out cross validation scheme (using the LOO [58] function). Crucially, using plausible (mildly informative) priors over random effects, this kind of analysis eludes a point null hypothesis—and allows us to specify, with a certain confidence, whether an effect was present or absent. This confidence is reflected by the 95% CrIs, as well as the posterior probability that a certain effect ($E\mu$) is different from 0 ($P(E\mu<0)$) or $P(E\mu>0)$). Consequently, for all relevant model parameters, we report expected values under the posterior distribution and their CrIs, as well as their posterior probabilities. In line with Franke and Roettger [59], we conclude that there is compelling evidence for an effect if its posterior probability $P(E\mu\neq0)$ is close to 1. We used generic weakly informative priors (in line with prior choice recommendations by the stan developer group; see https://github.com/stan-dev/stan/wiki/Prior-Choice-Recommendations), following a normal distribution for the intercept and all regression coefficients and a half-Cauchy distribution for residual and random effect variances (all prior distributions centred at 0). Each model was run for 4 sampling chains with a minimum of 5,000 iterations each (1,000 warm-up iterations). There were no indications of nonconvergence (all Rhat values = 1, no divergent transitions). All models discussed in this paper are listed in detail in the Supporting information.

The following Results section including all relevant data is publicly available as a reproducible R Markdown script at https://osf.io/xm7ty/.

## Results

### Haloperidol resulted in reduced labelling accuracy for both mental and non-mental state animations

A Bayesian mixed effects model (Model 1.1) with random intercepts for *subject ID* and *animation ID* (unique identifier for each animation) and a random slope for the effect of drug varying by subject ID was fitted to *accuracy* (see **Animations task**) and the dummy-coded predictor *drug* (haloperidol [HAL] versus placebo [PLA]; reference level = PLA; see Model 1, S1A–S1E Table). The model revealed a robust main effect of drug, where haloperidol resulted in lower accuracy in labelling the animations ($E\mu_{HALvsPLA} = -0.56$, CrI = [−0.94, −0.19]). The posterior probability that there was a truly negative effect ($P(E\mu_{HALvsPLA}<0)$ was 1. To further assess whether the drug specifically affected performance for mental state animations, the dummy-coded factor *mental state* (mental versus non-mental; reference level = non-mental), as well as the 2-way interaction between drug and mental state, was added to the model. This second model (Model 1.2) showed no interaction between drug and mental state ($E\mu_{HALvsPLA,mentalVSnonmental} = 0.20$, CrI = [−0.34, 0.74]), indicating that haloperidol decreased attribution accuracy to a comparable extent for mental and non-mental state animations. Furthermore, adding mental state to the model led to an even stronger effect of drug ($E\mu_{HALvsPLA} = -0.66$, CrI = [−1.11, −0.20]). Thus, after taking the drug, participants' ability to correctly classify an animation decreased by 0.66 points compared to the placebo condition (see Fig 1). These results, indicating a comparable influence of haloperidol on inference about

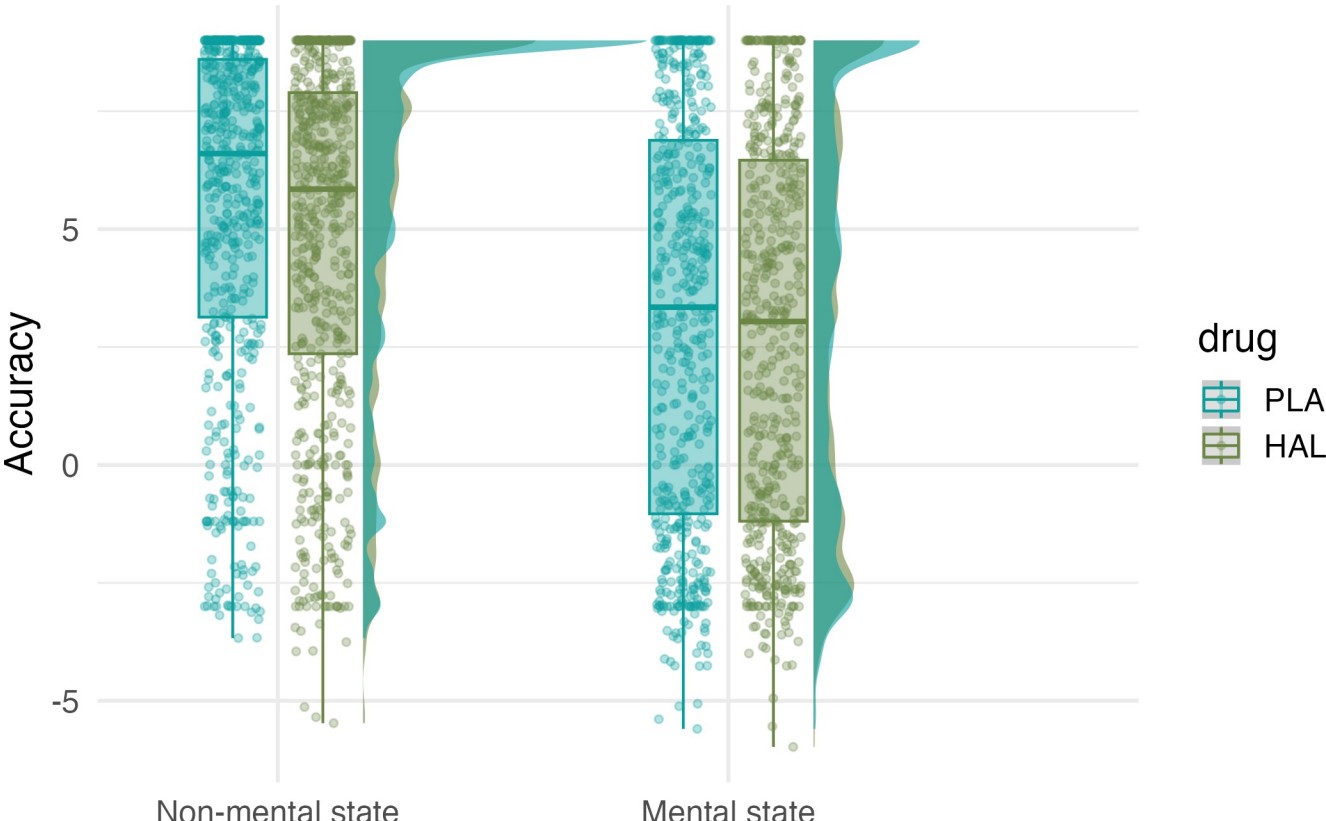

**Fig 1. Drug effects on accuracy by mental state condition.** HAL, haloperidol; PLA, placebo. Central marks of box plots correspond to the median; outer hinges correspond to the first and third quartiles (25th and 75th percentiles). Upper and lower whiskers extend to largest and lowest values at most 1.5 * IQR of the hinge. Data and code required to reproduce this figure available at https://osf.io/xm7ty/.

mental and non-mental states, may be mediated by dopamine's influence on general cognitive functions—such as working memory and attention—which play a key role in inferential reasoning [60]. We return to this question in our exploratory analyses after first testing our second hypothesis that dopamine may affect mentalising indirectly via its effects on movement. Finally, in line with our previous findings [30], a main effect of mental state ($E\mu_{mentalVSnonmental}$ = −2.50, CrI = [−3.21, −1.78]) suggests that overall, participants struggled more with interpreting animations depicting mental state interactions relative to ones displaying non-mental state interactions.

Control analyses: First, while model residuals did not violate the normality assumption of linear regression, visual inspection of the response variable revealed bimodality of our data. We confirmed the present results remain after this bimodality is taken into account by additionally modelling the response as a mixture of 2 gaussian distributions (see S1 Results, S1 Fig, and S5A–S5D Table). Second, to further investigate possible confounding effects of the day the drug was taken as well as potential effects of haloperidol on arousal levels, 2 control models were performed. Model 1.3 was fit to drug and drug day (day 1 versus day 2, dummy coded), as well as their interaction, predicting accuracy. There was no interaction between drug and drug day, indicated by a negative effect of drug for drug day 1 ($E\mu_{HALvsPLA,day1}$ = −0.52, CrI = [−1.07, 0.03]; note there is slightly increased uncertainty around the drug effect in this model, as shown by the CrI including 0), which did not differ from the drug effect on day 2 ($E\mu_{HALvsPLA,day1vsday2}$ = −0.06, CrI = [−0.81, 0.69]). Model 1.4 was fit to drug and *arousal* (participant reported tiredness levels ranging from 1 = not tired at all to 10 = maximally tired;

collected before the main task) predicting accuracy. The model revealed a preserved effect for drug ($E\mu_{HALvsPLA} = -0.52$, CrI = [−1.11, 0.07], again with minimal increase in uncertainty surrounding the effect) and no main effects for arousal (see S1D Table). There was an interaction between drug and a seventh order polynomial trend for arousal; however, model comparison between models 1.1 and 1.4 confirmed that arousal level did not meaningfully contribute to explaining variance in accuracy; we therefore did not interpret this result any further.

## Dopamine manipulation diminished the effect of movement similarity for mental state animations

To assess the contribution of dopamine disruption to the extent to which individuals make use of their own motor codes when judging the observed movements, *jerk difference* was calculated for both PLA and HAL trials by first subtracting the mean jerk (jerk was calculated as the third order non-null derivative of the raw positional data; for more details, see [30]) of each video a participant rated from their own jerk values when animating the same word, and then taking the absolute magnitude of those values. Thus, jerk difference indexes observer-animator movement similarity wherein lower values reflect higher jerk similarity. Subsequently, jerk difference was added to the previous model of drug and mental state (Model 1.2) predicting animations task accuracy. This new model (Model 2.1) reproduced the previous main effects of drug ($E\mu_{HALvsPLA} = -0.69$, CrI = [−1.14, −0.24]) and mental state ($E\mu_{mentalVSnonmental} = -2.71$, CrI = [−3.42, −1.99]). Furthermore, the model revealed an interaction between drug, jerk difference, and mental state, indicating that while under placebo, there was a stronger negative effect of jerk difference for mental, relative to non-mental state animations ($E\mu_{jerkDiff,non-mental,PLA} = -0.11$, CrI = [−0.37, 0.14]; $E\mu_{jerkDiff,mentalVSnonmental,PLA} = -0.54$, CrI = [−1.09, 0.00]), under haloperidol, this negative effect was diminished ($E\mu_{jerkDiff,mentalVSnonmental,HALvsPLA} = 0.68$, CrI = [−0.12, 1.47], $P(E\mu_{jerkDiff,HALvsPLA,mentalVSnonmental} > 0) = 0.95$; contrasts of jerk difference slope PLA versus HAL—non-mental state: $E\mu = -0.06$, CrI = [−0.37, 0.28], mental state: $E\mu = -0.74$, CrI = [−1.47, −0.01]). Separate post hoc models for placebo and haloperidol trials confirmed this pattern, with a robust negative effect of jerk difference for mental, but not non-mental state animations in the placebo model (Model 2.2: $E\mu_{PLA,jerkDiff,non-mental} = -0.13$, CrI = [0.41, 0.15]; $E\mu_{PLA,jerkDiff,mentalVSnonmental} = -0.69$, CrI = [−1.32, −0.08]), and no effect of jerk difference in either mental state condition in the haloperidol model (Model 2.3: $E\mu_{HAL,jerkDiff,non-mental} = -0.10$, CrI = [−0.38, 0.19]; $E\mu_{HAL,jerkDiff,mentalVSnonmental} = 0.02$, CrI = [−0.69, 0.72]). Consequently, under placebo, the higher the difference in jerk between an observer and the original animator of a given mental state animation, the less accurate the observer was in classifying that animation. Thus, the present placebo results are in line with our previous findings [30], this time emphasising a role for movement similarity in promoting inference from *mental* state animations. In contrast, under haloperidol, there was no such effect of movement similarity for the non-mental or the mental state animations (see Fig 2A and 2B).

The disappearance of the jerk difference effect in HAL trials suggests that under haloperidol, the relationship between one's own kinematics and the kinematics present in an animation stimulus did not affect accuracy. This result affords various interpretations. For example, it could be that participants rely less on their own motor codes (perhaps relying more on other sources of information such as visual codes) when judging animations under HAL. Alternatively, participants may rely on their own motor codes to an equal extent under HAL and PLA, but under HAL, they are relying on stored motor codes acquired across the lifetime, and upon which sensorimotor internal models are fine-tuned (i.e., their placebo movements), rather than those modified codes via which they are currently performing action [32,61]. To test the hypothesis that when observing the animations, individuals recruited their lifetime,

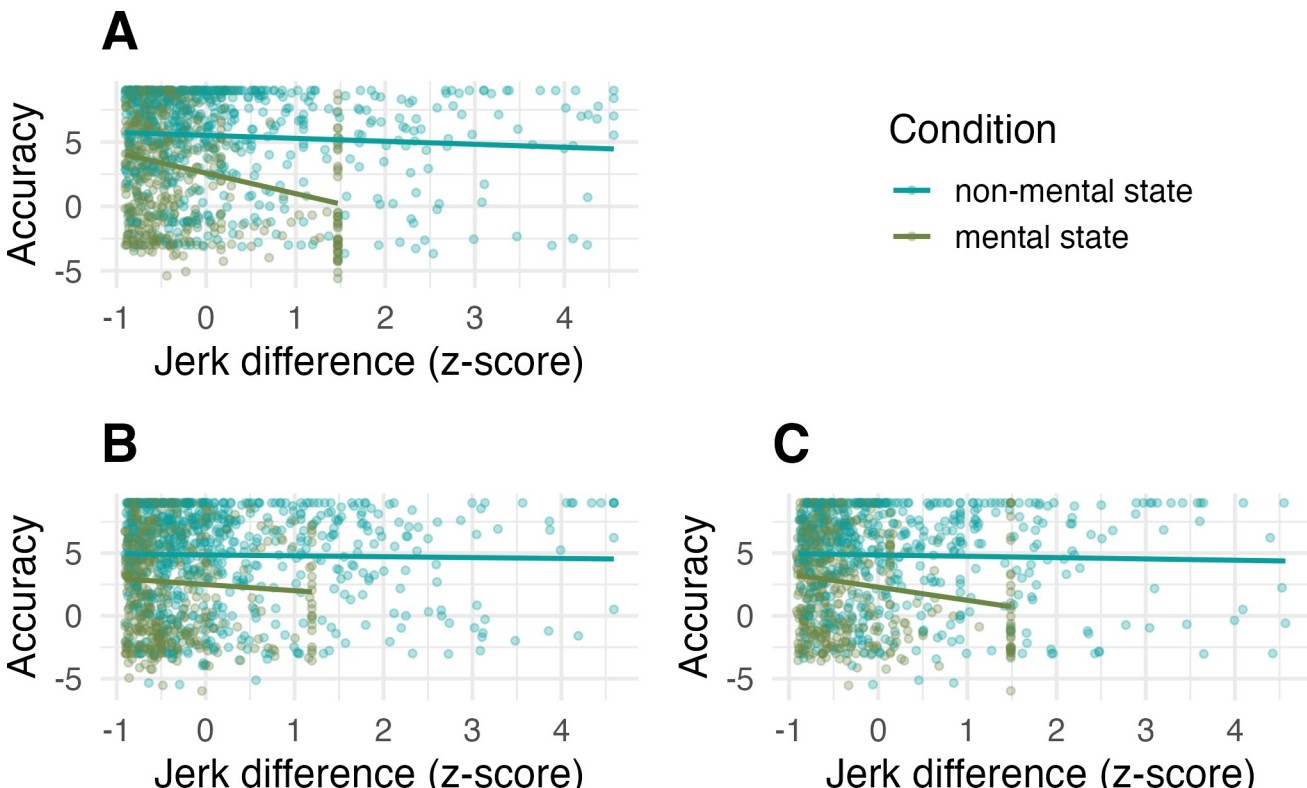

**Fig 2. Relationship between jerk difference and accuracy depends on how jerk difference was calculated.** (**A**) Placebo condition. (**B**) Haloperidol condition, jerk difference based on movement in haloperidol condition. (**C**) Haloperidol condition, jerk difference based on movement in placebo condition. Data and code required to reproduce this figure available at https://osf.io/xm7ty/.

experience-based motor codes under both PLA and HAL conditions, we calculated *placebo jerk difference* for each animation stimulus viewed in the haloperidol condition by subtracting a given animation stimulus' mean jerk from the participant's own jerk in the placebo condition.

A new model (Model 3.1) with placebo jerk difference added as covariate revealed the same main effect of drug ($E\mu_{HALvsPLA}$ = −0.73, CrI = [−1.22, −0.24]), as well as the same interaction between jerk difference and mental state as before ($E\mu_{jerkDiff,non-mental}$ = −0.16, CrI = [−0.41, 0.10]; $E\mu_{jerkDiff,mentalVSnonmental}$ = −0.55, CrI = [−1.11, 0.00]). However, there was no interaction between jerk difference, mental state, and drug ($E\mu_{jerkDiff,HALvsPLA,mentalVSnonmental}$ = 0.20, CrI = [−0.54, 0.94]), indicating a negative effect of placebo jerk difference for mental state animations for both PLA and HAL trials (see Fig 2C). To further corroborate this finding that placebo, and not haloperidol jerk difference affected accuracy in both PLA and HAL trials, we ran a second model (Model 3.2) with only HAL trials and both placebo and haloperidol jerk difference as predictors. This model clearly showed no effect of haloperidol jerk difference ($E\mu_{HALjerkDiff,non-mental}$ = 0.03, CrI = [−0.35, 0.40]; $E\mu_{HALjerkDiff,mentalVSnonmental}$ = −0.30, CrI = [−1.14, 0.50]), and an even stronger effect of placebo jerk difference ($E\mu_{PLAjerkDiff,non-mental}$ = −0.06, CrI = [−0.44, 0.31]; $E\mu_{PLAjerkDiff,mentalVSnonmental}$ = −0.70, CrI = [−1.38, −0.03], $P(E\mu_{PLAjerkDiff,mentalVSnonmental} < 0)$ = 0.98, coefficient for placebo jerk difference in mental state animations = −0.06 + (−0.70) = −0.76) on mental state animations. In other words, when participants were labelling animations under haloperidol, accuracy in those judgements was affected by their own movements produced in the placebo, but not by their movements produced in the drug condition. Models 2.1 and 3 show that under both HAL and PLA,

participants' accuracy on the animations task is influenced by the similarity of the animation to the movements that they produced in the placebo condition.

## Effects of haloperidol on animations task accuracy show specific relationships to drug effects on emotion recognition

To probe potential underlying mechanisms of the observed drug effects on accuracy in the animations task, we investigated relationships between drug-related changes in animations task accuracy and drug effects on tasks indexing emotion recognition and executive functions. For this, we first created a variable that indexed drug effects on animations task accuracy on an individual participant basis. Due to the random selection of animations (see [30]), participants did not necessarily view the same animations in PLA and HAL trials, making it impossible to calculate a trial-by-trial measure of drug-related changes. Thus, the accuracy measure was first transformed into a binary variable, classifying as "correct" any trial where the highest rating was given to the target word, while a trial where the highest rating was given to a non-target word was classed as "incorrect." Subsequently, the percentage of correct trials out of all 8 trials for a given word was calculated, resulting in 2 *percentage accuracy* values per animation word per participant (1 PLA, 1 HAL). Finally, for each participant, drug-related changes in accuracy were calculated by subtracting percentage accuracy values of placebo days from those collected on drug days. *Animations task accuracy change* scores, therefore, represent the change in percentage of correct trials from placebo to haloperidol conditions, whereby positive values indicate enhanced ability to correctly label the animations after the drug, and negative values indicate a decrease in labelling accuracy.

While there is a relatively large evidence base implicating dopaminergic signalling in general cognition, including executive function [62] and learning [63], the literature is less conclusive about the role of dopamine in *socio*cognitive processes. Thus, to investigate whether our observed drug effects on animations task accuracy were related to drug effects on sociocognitive performance above and beyond expected relationships with drug effects on executive functions, we calculated *working memory change* scores as index of drug effects on working memory span and *emotion recognition change* scores indexing drug effects on emotion recognition by subtracting accuracy scores of PLA trials from those obtained in HAL trials for both tasks. For both indices, positive change scores indicate increased performance under haloperidol. If we observed specific relationships between drug-related changes in emotion recognition and changes in accuracy for mental state, but not non-mental state animations, this would provide support for specific effects of dopamine challenge on sociocognitive processes. A Bayesian linear model (Model 4.1; model comparison revealed that random intercepts for subject ID did not additionally explain variance; see S4B Table) was fit to emotion recognition change, working memory change, and mental state (mental, non-mental; dummy-coded, reference level = non-mental) as well as interactions between mental state and both continuous predictors, predicting animations task accuracy change. The discrete response variable was modelled as a student's t distribution (a continuous was chosen over a cumulative model distribution based on model comparison using LOO [58] showing clear preference for the discretised continuous model; for more details, see S4C Table). The first model revealed no effect for working memory change ($E\mu_{WMchange,non-mental}$ = −0.00, CrI = [−0.01, 0.00]; $E\mu_{WMchange,mentalVSnonmental}$ = −0.00, CrI = [−0.02, 0.00]); hence, all subsequent effects are reported based on a model excluding this variable (Model 4.2). Model 4.2 revealed an interaction between emotion recognition change and mental state, with no relationship between emotion recognition change and animations task accuracy change for non-mental state animations ($E\mu_{ERchange,non-mental}$ = −0.02, CrI = [−0.07, 0.03]) and, relative to non-mental state animations, a small effect indicating a positive

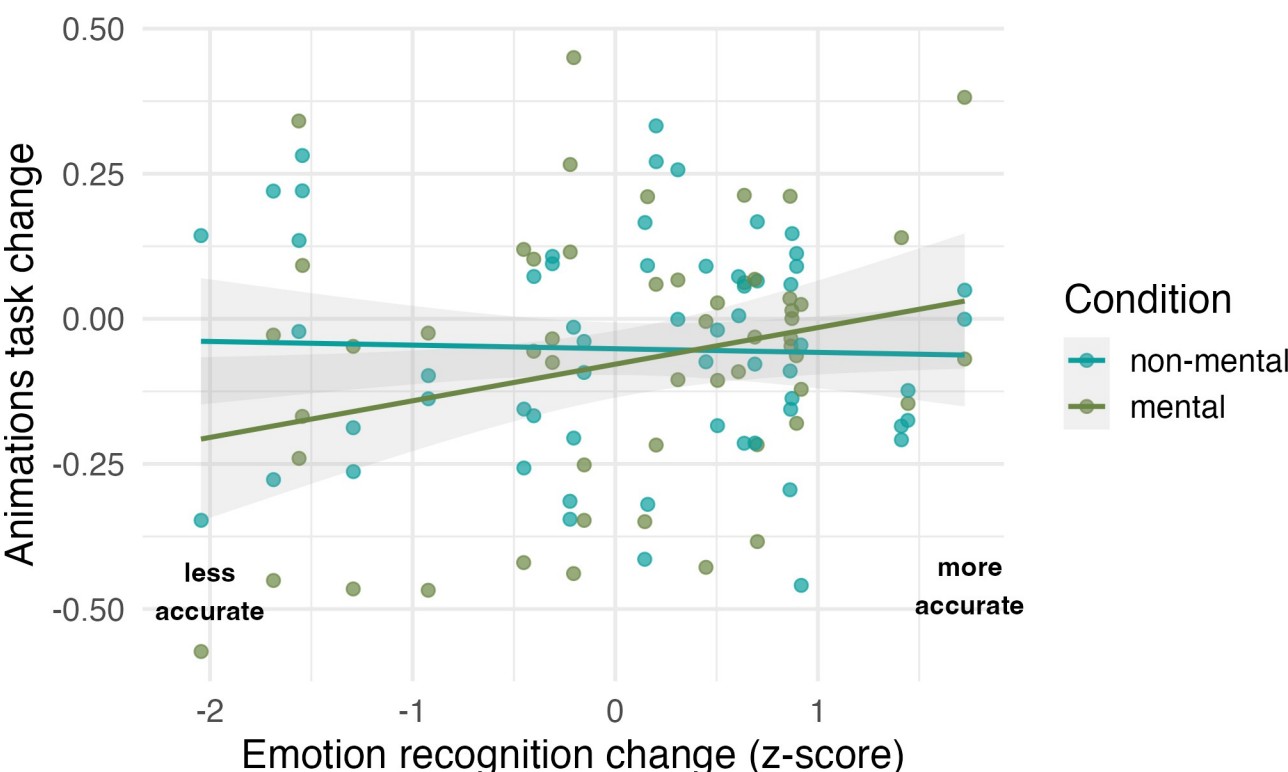

**Fig 3. Relationship between drug effects on animations task performance and emotion recognition performance.** Vertical jitter was added to the raw data points for display purposes. Data and code required to reproduce this figure available at https://osf.io/xm7ty/.

relationship between mental state accuracy and ER accuracy (Fig 3; $E\mu_{ERchange,mentalVSnonmental}$ = 0.07, CrI = [−0.00, 0.13]; $P(E\mu_{ERchange,mentalVSnonmental}>0)$ = 0.96; coefficient for mental state animations $E\mu_{ERchange,mental}$: −0.02 + 0.07 = 0.05; note the CrI including 0 indicates some uncertainty surrounding this effect). Thus, for every 2 SD increase in emotion recognition accuracy after haloperidol, individuals correctly identified roughly 1 animation more (0.05 * 2 * 8 = 0.8) than under placebo, relative to those participants who showed no change in emotion recognition performance. Finally, there was no main effect of mental state ($E\mu_{mentalVSnonmental}$ = −0.01, CrI = [−0.08, 0.06]), further confirming our results from model 1 that the drug affected performance equally for mental and non-mental state animations.

## Discussion

The present study used pharmacological challenge of dopamine function in combination with a classical mentalising task to evaluate whether dopamine is causally implicated in mental state attribution. Our secondary aim was to probe mechanistic pathways involved in the dopaminergic modulation of mental state attribution. To this end, we investigated relationships between effects of the dopamine manipulation on the main mentalising task and tasks indexing working memory and emotion recognition. To our knowledge, this is the first study to show detrimental effects of pharmacological dopamine manipulation on mentalising ability in a sample of healthy adults. More precisely, individuals showed reduced ability to adequately label mental state animations after administration of the dopamine D2/D3 antagonist haloperidol compared to placebo, indicating a causal role for dopamine in mental state attribution. Our findings thus show that dopaminergic pathways are, at least indirectly, involved in ToM.

Furthermore, our data did not show an interaction between the dopamine manipulation and the type of animation, indicating that haloperidol affected participants' performance for mental state and non-mental state animations to a comparable extent. Importantly, however, the results of our secondary analyses suggest that dopamine may modulate inferences from mental and non-mental state interactions via separate pathways. We observed that effects of haloperidol on participants' ability to correctly identify mental, but not non-mental, state animations were positively related to drug effects on their emotion recognition performance. In other words, individuals who exhibited a decrease in their mentalising ability as a response to dopamine antagonism were more likely to also have shown reduced emotion recognition performance in a dynamic whole-body emotion perception task. In contrast, drug effects on inferences from non-mental state animations were unrelated to drug-induced impairments in emotion recognition. Furthermore, effects of haloperidol on individuals' working memory capacity did not contribute to explaining drug-related impairments in mental state attribution.

We thus speculate that these results suggest a common dopamine modulated mechanism among mental state attribution and emotion recognition ability that is independent of working memory function as indexed by the Sternberg [41] visual WM task. Recent empirical evidence from animal [64,65] and clinical [66,67] studies supports the idea that dopamine may modulate social behaviour via reward-related mechanisms. In the context of the current study, a haloperidol-induced decrease in dopamine transmission in the mesolimbic dopamine pathway—a core brain circuit for processing reward [68]—may have resulted in participants failing to pick up specific social cues from the animations. Alternatively, recent research suggests that haloperidol may have affected the processing of social cues via coding for their perceived self-relevance: For instance, in a simple dictator game, haloperidol resulted in a shift in intentional attributions to a partner's behaviour along an axis of self-relevance, leading to a reduction in attributions of harmful intent (i.e., relevant to the participant as harmful intent represents threat) alongside an increase in attributions of self-interest (not relevant to the participant) [16]. While self-relevance has been shown to impact stimulus processing at various stages (e.g., visual, attention, memory [69,70]), it is yet unclear whether self-relevance and reward effects rely on shared or distinct mechanisms [71]. Future research is needed to confirm the exact mechanistic and neural pathway(s) underlying the dopaminergic modulation of mentalising and emotion recognition.

We further hypothesised that dopamine challenge may affect participants' mentalising ability by impeding their capacity to internally represent observed movements. Crucially, our data provide a partial replication of our previous results [30], showing that under placebo, movement similarity between observer and animator promotes inference from mental state animations. The present results are thus consistent with action simulation accounts [25,28,72], which suggest that individuals implicitly map observed actions onto their own motor system and that this can influence how they label observed actions. Moreover, the selectivity of the movement similarity effect to *mental* state animations suggests that our movement similarity measure (i.e., jerk difference) may be indexing mapping at higher levels of the "motor hierarchy" [46], reflecting integration of the motor action with its underlying intention (i.e., mentalising), rather than mapping of short-term action goals (action understanding). Within the putative mirror neuron network, the inferior parietal lobule [73,74] may be a candidate node for these higher-order action representation processes, as this region is causally implicated in decoding intentions from action kinematics [75] and recruited during animations tasks [76] (i.e., responds to kinematics without the presence of body parts). Intriguingly, we observed that under haloperidol, individuals showed the same movement similarity effect on mentalising when movement similarity was calculated based on their movements produced in the placebo condition, but not when the measure was based on movements from the haloperidol condition.

These findings, alongside the observation that haloperidol acutely affected individuals' motor function (see S2 Results), give rise to the idea that sensorimotor representations built from a lifetime of visual and motor experiences are robust against short-term disruptions of motor output. More precisely, while haloperidol acutely affected participants' ongoing movements, our data suggest that their internal motor codes associated with the relevant mental states were unaffected; i.e., during mentalising under dopamine challenge, participants may have recruited visuo-motor representations that were formed before their actions were affected by the drug (i.e., through their lifetimes' experience of associating visuo-motor representations with mental state labels). In support of this explanation, we observed that, on the drug day, participants were more accurate in labelling mental states for animations that were kinematically similar to their own movements on the placebo day. This hypothesis is consistent with studies of PwP that suggest that in early stages of the disease, during action observation, PwP recruit action representations developed during the presymptomatic stage; it is only at later stages that these representations change due to the increasing severity of motor symptoms ([9,47,77]; although see [23]). Ultimately, while sensorimotor representations of mental states may still be intact in the early stages of PD, our results show that deficits in ToM functioning can occur even after very short-term perturbations of dopamine transmission, presumably due to the disruption of mechanisms unrelated to action representation processes.

There are some limitations to the conclusions that can be drawn from the current study. First, while animations tasks have been widely used to probe sociocognitive processes due to their ability to reliably distinguish between clinical and control groups [30], it can be argued that their ecological validity is limited compared to experimental designs involving realistic social interactions. Real-world social situations give rise to a multitude of clues to others' mental states, including facial expressions, tone of voice, and content of verbalisations, and involve repeated interactions—all of which the present task is not designed to index. While the dopaminergic modulation of mentalising during recursive social interactions has been investigated, for instance, using multiround economic games [16,17], future work could expand on the existing results by investigating how dopamine modulates the attribution of mental states to human agents in face-to-face interactions. Moreover, while previous work suggests that dopamine may modulate mental state *representation* (given its role in model-based reasoning [78]), based on the current data, it cannot be determined whether performance differences are driven by differences in the representation, or mere inference of those mental states [79]. Second, our interpretation that the observed effects of our dopaminergic manipulation in the mental and non-mental state conditions arise from 2 separable neurochemical (i.e., mesolimbic and nigrostriatal) and mechanistic (action representation and reward processing) pathways is based on subsidiary analyses and warrants further investigation using dedicated experimental designs. Computational psychopharmacology and pharmacological fMRI/PET both offer fruitful routes to expanding our current understanding of how dopamine regulates sociocognitive processes. Third, while our results suggest that haloperidol affected mentalising performance independent of working memory function, this does not preclude other aspects of executive function, such as attention, inhibitory control, or cognitive flexibility [80] playing a role in the dopaminergic modulation of mentalising. Finally, dopamine is likely not acting on social function in isolation. Although dopaminergic treatment may benefit sociocognitive function (potentially by reducing suboptimally high dopamine action to more optimal levels), there is growing evidence that the dopaminergic system may work in interaction with the serotonergic system, evidenced, for instance, by reports of therapeutic effects of atypical (acting on both dopaminergic and serotonergic receptors), but not classical (targeting specifically dopamine receptors), antipsychotics on ToM [12,24,81]. Future work is needed to identify the specific contributions of the dopaminergic and serotonergic, and other neuromodulatory systems, to sociocognitive function.

In conclusion, our data causally implicate D2/D3 dopamine in mentalising. Our secondary findings highlight 2 putative pathways via which dopamine disruptions may affect mentalising ability. The present study thus adds further support to a line of research [11,12,82] indicating the potential of dopaminergic treatment in sociocognitive dysfunction, calling to attention the need for further research into the exact neurochemical and computational bases of the dopaminergic modulation of mental state attribution.

## Supporting information

**S1 Fig. Results of gaussian mixture model.** HAL, haloperidol trials; PLA, placebo trials. (**A**) Probability density plot of the response variable accuracy. (**B**) Posterior probability distribution of model 1.2. Y = response distribution, $y_{rep}$ = 100 draws from posterior samples. (**C**-**F**) Conditional effects plots (created using the "conditional_effects" function of the brms package [55]) for simple gaussian models fit to each distribution component individually, (**C**, **D**) depicting the interaction term of drug and mental state, (**E**, **F**) depicting the interaction term of drug and baseline WM. C = distribution component 1 comprising accuracy values < 3.01, D = distribution component 2 with accuracy values > 3.01. E = distribution component 1 comprising accuracy values < 3.01, E = distribution component 2 with accuracy values > 3.01. (DOCX)

**S1 Tables. S1A Table**. **Model parameters for model 1.1.** Model formula: accuracy ~ drug + (1 + drug || subject ID) + (1 | animation ID). **S1B Table**. **Model parameters for model 1.2.** Model formula: accuracy ~ drug * mental state + (1 + drug || subject ID) + (1 | animation ID). **S1C Table**. **Model parameters for model 1.3.** Model formula: accuracy ~ drug * drug day + (1 + drug || subject ID) + (1 | animation ID). **S1D Table**. **Model parameters for model 1.4.** Model formula: accuracy ~ drug * arousal + (1 + drug || subject ID) + (1 | animation ID). L = linear-, Q = quadratic-, C = cubic-, E4-E7 = fourth-seventh order polynomial trend. **S1E Table**. **Leave-one-out (Loo) cross-comparison of models 1.1 and 1.4.** Elpd_diff = Bayesian LOO estimate of the expected log pointwise predictive density (see [58]); se_diff = standard error of elpd_diff. Model weights were obtained using the brms function "model_weights." (DOCX)

**S2 Tables. S2A Table**. **Model parameters for model 2.1.** Model formula: accuracy ~ drug * mental state * jerk difference + (1 + drug || subject ID) + (1 | animation ID). **S2B Table**. **Model parameters for model 2.2 (PLA only).** Model formula: accuracy ~ jerk difference * mental state + (1 | subject ID) + (1 | animation ID). **S2C Table**. **Model parameters for model 2.3 (HAL only).** Model formula: accuracy ~ jerk difference * mental state + (1 | subject ID) + (1 | animation ID). (DOCX)

**S3 Tables. S3A Table**. **Model parameters for model 3.1.** Model formula: accuracy ~ drug * mental state * PLA jerk difference + (1 + drug || subject ID) + (1 | animation ID). **S3B Table**. **Model parameters for model 3.2.** Model formula: accuracy ~ mental state * PLA jerk difference * HAL jerk difference + (1 + subject ID) + (1 | animation ID). (DOCX)

**S4 Tables. S4A Table**. **Model parameters for model 4.1.** Model formula: accuracy change | trunc(lb = −1, ub = 1) ~ emotion change * mental state + WM change * mental state. ER change = emotion recognition change; WM change = working memory change. **S4B Table**. **Leave-one-out (loo) cross-comparison of models 4.1 and 4.1.rand.** Elpd_diff = Bayesian LOO estimate of the expected log pointwise predictive density (see [58]); se_diff = standard

error of elpd_diff. Model weights were obtained using the brms function "model_weights."
**S4C Table**. **Leave-one-out (loo) cross-comparison of models 4.1 and 4.1.cum.**
Elpd_diff = Bayesian LOO estimate of the expected log pointwise predictive density (see [58]);
se_diff = standard error of elpd_diff. **S4D Table**. **Model parameters for model 4.2**. Model formula: accuracy change | trunc(lb = −1, ub = 1) ~ ER change * mental state. ER
change = emotion recognition change.
(DOCX)

**S5 Tables**. **S5A Table**. **Model parameters for model 5.** Model formula: accuracy ~ drug *
mental state + (1 | subject ID). Response modelled as a mixture of 2 gaussian distributions.
**S5B Table**. **Model parameters for model 6.1.** Model formula: accuracy ~ drug * WM + (1 |
subject ID). WM = working memory; response modelled as a mixture of 2 gaussian distributions. **S5C Table. Model parameters for model 6.2 (post hoc model—low WM).** Model formula: accuracy ~ drug + (1 | subject ID). Response modelled as a mixture of 2 gaussian
distributions. **S5D Table**. **Model parameters for model 6.3 (post hoc model—high WM).**
Model formula: accuracy ~ drug + (1 | subject ID). Response modelled as a mixture of 2 gaussian distributions.
(DOCX)

**S6 Tables**. **S6A Table**. **Model parameters for model 7.1.** Model formula: speed ~ drug * WM
+ (1 | subject ID). **S6B Table**. **Model parameters for model 7.2 (post hoc model—low WM).**
Model formula: speed ~ drug * WM + (1 | subject ID).
(DOCX)

**S1 Text. Eligibility criteria.**
(DOCX)

**S1 Results. Modelling the bimodality of the response.**
(DOCX)

**S2 Results. Dopamine challenge reduced walking speed in individuals with low estimated
dopamine synthesis capacity.**
(DOCX)

## Acknowledgments

We would like to thank Lukas Lengersdorff for his statistical advice.

## Author Contributions

**Conceptualization:** Bianca A. Schuster, Sophie Sowden, Alicia J. Rybicki, Jennifer L. Cook.

**Data curation:** Bianca A. Schuster.

**Formal analysis:** Bianca A. Schuster, Dagmar S. Fraser.

**Funding acquisition:** Jennifer L. Cook.

**Investigation:** Bianca A. Schuster, Sophie Sowden, Alicia J. Rybicki, Lydia Hickman.

**Methodology:** Bianca A. Schuster, Clare Press, Jennifer L. Cook.

**Project administration:** Bianca A. Schuster, Sophie Sowden, Alicia J. Rybicki, Jennifer L. Cook.

**Resources:** Peter Holland.

**Writing – original draft:** Bianca A. Schuster.

**Writing – review & editing:** Bianca A. Schuster, Sophie Sowden, Alicia J. Rybicki, Dagmar S. Fraser, Clare Press, Lydia Hickman, Jennifer L. Cook.

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
