## [Editor Report · Decision Letter 0]

29 Aug 2023

Dear Dr Schuster, 

Thank you for submitting your manuscript entitled "Dopamine challenge reduces mental state attribution accuracy" for consideration as a Research Article by PLOS Biology.

Your manuscript has now been evaluated by the PLOS Biology editorial staff as well as by an academic editor with relevant expertise and I am writing to let you know that we would like to send your submission out for external peer review.

We also noted that there is no information about pre-registration provided in your study. Since this information is available in your previous eLife paper, we were wondering if you could include this information also in the current manuscript.

Once your full submission is complete, your paper will undergo a series of checks in preparation for peer review. After your manuscript has passed the checks it will be sent out for review. To provide the metadata for your submission, please Login to Editorial Manager (https://www.editorialmanager.com/pbiology) within two working days, i.e. by Aug 31 2023 11:59PM.

Kind regards,

Christian

Christian Schnell, PhD

Senior Editor

PLOS Biology

cschnell@plos.org

---

## [Decision Letter · Decision Letter 1]

23 Oct 2023

Dear Dr Schuster,

Thank you for your patience while your manuscript "Dopamine challenge reduces mental state attribution accuracy" was peer-reviewed at PLOS Biology. It has now been evaluated by the PLOS Biology editors, an Academic Editor with relevant expertise, and by several independent reviewers. 

In light of the reviews, which you will find at the end of this email, we would like to invite you to revise the work to thoroughly address the reviewers' reports.

As you will see below, the reviewers find your study very interesting but raise a couple of concerns on different aspects of your study. In your revision, we would in particular expect you to address the concerns about the discrepancies in model 2 and the reported Bayesian statistics, to improve lack of clarity of the results presented in Figure 3, and to add a new random intercept analysis for the comparison between haloperidol and placebo conditions. Furthermore, please tone down the conclusions regarding the link between the animation tests and Theory of Mind, whether non-mental state and mental state attributions are made with any discernible reasoning, and whether the control tasks that you used are adequate enough to draw conclusions about effects on emotional recognition and motor control.

We also ask you to carefully look into the Bayesian statistics, as we are likely to add an additional reviewers in the next round who can assess this aspect of your study.

Given the extent of revision needed, we cannot make a decision about publication until we have seen the revised manuscript and your response to the reviewers' comments. Your revised manuscript is likely to be sent for further evaluation by all or a subset of the reviewers.

**IMPORTANT - SUBMITTING YOUR REVISION**

*Re-submission Checklist*

*Published Peer Review*

*PLOS Data Policy*

*Blot and Gel Data Policy*

Sincerely,

Christian

Christian Schnell, PhD

Senior Editor

PLOS Biology

cschnell@plos.org

REVIEWS:

Reviewer's Responses to Questions

Reviewer #1: This was an well-written, clear, and interesting paper reporting the effects of a D2 DA challenge on theory of mind performance in healthy adults. I am not an expert in either pharmacology nor kinematic analysis, though, I do know about much of the work that has attempted (with varying degrees of success) to connect DA functioning more broadly with theory of mind. 

The authors use a relatively novel task within a convincing design (double-blind placebo) to show that DA challenge affects labeling of animations that require mentalizing. The DA challenge also affects the labeling of animations that don't require mentalizing. The effect of DA challenge on mentalizing on non-mentalizing was not statistically distinguishable in the main task. Yet they conclude that DA has a specific effect on mentalizing. This conclusion is based on subsidiary analyses that I didn't think provided empirical support commensurate with the strength of the conclusions. 

It was unclear to me what the kinematic analyses showed with respect to the main research question — the authors previously showed that the similarity that one's own movements show to a model's movements predict their accuracy on the mentalizing animations. This effect goes away where in the DA challenge, but they speculate that this is because movements under DA challenge are odd for participants. They show that movement similarity under placebo predicts mentalizing accuracy under DA challenge, but I struggled to see how this shed light on the connection between DA and animation labeling accuracy. I gather that the authors are trying to argue that even though DA affects movement, it's not the effects on movement that are directly affecting their ToM accuracy. But this seems like argument from null and is pretty weak. 

More interesting was that DA challenge also affected emotion recognition accuracy in a separate task, and the extent to which it did so for a given participant was weakly correlated with their decrement in performance on the mentalizing animations. The authors speculate that this is evidence for a pathway whereby DA affects social cognition independent of any effects that it might have on other aspects of cognition — they single out executive function as an important example. Perhaps I don't know enough about the movement measures, but since EF is not measured here, it's not clear how this speculation is supported. Do we know that EF is not important for the emotion recognition task used here (I liked, but was new to me), or just not for other emotion recognition tasks generally? 

In general, I found these data very interesting, but the strong conclusions not yet warranted. 

A minor consideration — there is a small amount of evidence that DA might be related to some aspects of children's theory of mind development (e.g., Lackner et al., 2010; Sabbagh et al., 2012), also independent of whatever effects DA might have on EF. Also, the Savina & Beninger paper comparing the effects of typical and atypical antipsychotics (cited near the end of this paper) might provide interesting evidence — taking haloperidol didn't increase ToM performance in this group, but taking clozapine and olanzapine did (without affecting control tasks). I don't know all the pharmacology here (except that I know it's complicated), but it seems like a dissociation worth mentioning in the intro trying to understand current state of the evidence on DA and ToM. 

Reviewer #2: This is an interesting study of the effects of a dopamine antagonist on the attribution of intention to moving shapes in healthy adults. The paper makes a case for a causal implication between dopamine and the theory of mind, and that dopaminergic modulation of mental and non-mental state attributions involve distinct pathways. 

There are clear strengths in the within-subject design, placebo control, and analysis. The links with motor responses were an interesting hypothesis. 

My three primary concerns are that the mental and non-mental state categorisation does not seem to be meaningful, dependent variables were used inconsistently, and a single working memory task is insufficient to rule out general effects on executive function and task performance, as claimed. In the least charitable interpretation, we see that haloperidol reduces computerised task performance, including deducing what might be intended in an animated triangles video and communicating that deduction. In the most charitable interpretation (with potential revisions), we see that a dopamine antagonist, directly or indirectly, may influence the ability to infer mental states, independent of a specific type of working memory. As discussed below, I would not endorse the exploratory analyses based on video categories at this time. 

For consideration: 

* Animation target words were: 'fighting, following, seducing and surprising'. The experimenters divided these four words into two categories: based on Schuster et al 2021, 'following' and 'fighting' as 'non-mental state' (because there is reciprocal interaction, but supposedly no implication that one character was reading the other triangle's 'mind') and 'seducing' and 'surprising' animations as 'mental state' attributions because one character reacts to the other character's mental state. There is nothing I could find in the paper to justify the classification. 

If the label 'mental state' attribution refers to the participant's inference of mental states, then participants could attribute a mental state to 'fighting' and 'following' to at least one, and potentially two triangles in the animation, but these are classified as non-mental state attributions. There is the intentionality of a triangle in each of these concepts, which the participant could infer. 

If the label 'mental state' attribution refers to inferring the triangle's inference of mental states, then it is unclear why that would be interesting. Even so, at least one and possibly two triangles are likely to infer the mental state of the other when fighting or following them. 

So, what I am trying to say is that an experimental distinction between these categories does not appear to be correct or meaningful. 

Without a detailed reading of the methods and a glance at Schuster et al. 2021, a reader could have assumed that 'non-mental state' attribution referred to a variant of the animation task where they moved randomly or in geometric patterns, which would perhaps have been more meaningful. So as a more minor point, the distinction between the conditions should be clearly communicated in the abstract, intro and discussion. These categories are not considered in the introduction at all. 

* Unfortunately, I do not agree with the experimenter's conclusion that these results suggest a common dopamine-modulated mechanism among mental state attribution and emotion recognition ability independent of executive function. A single working memory task is insufficient to measure executive function to make any inferences beyond that type of working memory. This is because executive function also involves response inhibition, interference control (e.g., selective attention), and cognitive flexibility. Diamond A. Executive functions. Annu Rev Psychol. 2013;64:135-68. 

* Line 324. It is unclear why the inability to compare PLA and HAL trials directly, due to conditions not experiencing the same videos, led to a choice for a very noisy binary measure of accuracy on the triangles task and a new measure of 'percentage accuracy'. With such a noisy measure, a trial in which the target word was rated 99% higher is weighted the same as a trial where it was rated 1% higher than the others. So that is not preferable and it is unclear how this solves the incomparability problem between conditions. In the main analysis, a random intercept was introduced for the animation ID. Why not do that here as well? The disconnect between this exploratory analysis and the main analysis is somewhat troubling and requires further explanation. 

* I worry that many of the credibility intervals overlap with zero (all the parameters for model 4.2 in Supp Table 4.3). Does this support the concrete inferences the authors are making? Figure 3 suggests there is not really a strong delineation between mental state attribution categories and correspondence between triangles task performance and emotion recognition performance. The reported stats written in the main text for model 4.2 do not match the stats reported in Supp Table 4.3, which does not reassure that model fitting was done correctly and selectively. I am not commenting further on the Bayesian stats, because they are beyond my expertise, so I hope another reviewer may offer their thoughts on them. They should have specialist (which I am not) reviewer attention before acceptance for publication.

Minor comments:

* Line 328 There are four words in each drug condition, so how do you end up with four percentage accuracy scores and not 8? Presumably, they developed percentage accuracy scores per category….. 

* Model results, in tables, may be more appropriate in the main text. 

* For the modelling (Line 376), it seems that random intercepts (subject IDs) may have not been included after deciding that they did not contribute to model fit. I would suggest this should be included regardless, as a minimum. 

* Line 395: Why are the units of increase in emotion recognition standard deviations, but units of increase of triangles accuracy in %? Why not standardise both? What does '5% increase in their ability' mean when the outcome is a binary variable? 

* Line 423-426: The author's phrasing implies a double dissociation, but it should be clear that this is not the case and that inferences should not be made from a negative result.

* The 'causal 'relation claim has been overstated, which could mistakenly be used as a cornerstone of a theory that dopamine directly modulates mentalising. The authors should be ve

---

## [Decision Letter · Decision Letter 2]

5 Mar 2024

Dear Dr Schuster,

Thank you for your patience while we considered your revised manuscript "Dopamine challenge reduces mental state attribution accuracy" for consideration as a Research Article at PLOS Biology. Your revised study has now been evaluated by the PLOS Biology editors, the Academic Editor and the original reviewers. Please not that Reviewer 1 did not re-review the manuscript but we recruited Reviewer 4 as an additional statistical reviewer. 

In light of the reviews, which you will find at the end of this email, we are pleased to offer you the opportunity to address the remaining points from the reviewers in a revision that we anticipate should not take you very long. We will then assess your revised manuscript and your response to the reviewers' comments with our Academic Editor aiming to avoid further rounds of peer-review, although might need to consult with the reviewers, depending on the nature of the revisions.

**IMPORTANT - SUBMITTING YOUR REVISION**

*Resubmission Checklist*

*Published Peer Review*

*PLOS Data Policy*

*Blot and Gel Data Policy*

Sincerely,

Christian

Christian Schnell, PhD

Senior Editor

PLOS Biology

cschnell@plos.org

REVIEWS:

Reviewer #2: Reviewer comments

The author responses to my earlier comments were excellent, and unless stated otherwise, I find their responses satisfactory. As mentioned before, the study's strengths lie in its within-subject design and robust human pharmacological methods. However, I still have some lingering concerns about the conclusions drawn in this paper. In this follow-up evaluation, I'll address the claims made by the authors in the order presented and the evidence supporting those claims. Some of this may overlap with my previous review if certain points weren't adequately addressed. 

Summary: 

Each conclusion presents its own set of challenges. The primary conclusion establishes that dopamine manipulation can impact mental state attribution across various animated scenarios, but it lacks significant insight into the underlying mechanism or specificity of effect for social cognition that would take us a significant step forward. On the other hand, the secondary conclusion relies on a questionable distinction between video categories without robust theoretical support for the tests and with questionable statistical inferences. I genuinely wish I could be more positive, but it's challenging to fully endorse the conclusions as they are.

More detail:

We still need a Bayesian's oversight on the confidence intervals question (but also the lack of correction for multiple comparisons). 

The primary finding here is the observed decrease in the accuracy of animations labeling with the administration of haloperidol, which suggests a connection to dopamine in Theory of Mind (ToM). After replicating this effect in a mixed model with the shared dataset, I do agree that the basic effect is indeed present. It's great that the authors made some qualifications in the discussion, which I appreciate.

However, the primary claim of a causal role in ToM still takes the spotlight in the abstract. It might be helpful to acknowledge that this assertion could potentially mislead readers, as we discussed in my previous review. Since there isn't enough evidence to directly link dopamine modulation to ToM, it's important to be cautious in interpreting the findings and adjust the language accordingly throughout the manuscript. For example, using phrasing in the abstract similar to the later statement on lines 437-439 "Our findings thus demonstrate that dopaminergic pathways impact [changed from involved in] Theory of Mind, at least indirectly," would more accurately reflect the scope of the evidence presented.

I also noticed that some labels in the provided dataset aren't very clear. It might be beneficial to include a glossary to explain terms like 'Halday' and 'drugday' to avoid any confusion, especially since their distinctions remain ambiguous despite differences in values, and accuracy differences aren't apparent in the datasets.

With these adjustments made throughout the manuscript, the conclusion becomes acceptable. However, standing alone, I'm not entirely convinced it meets the criteria of exceptional importance and interest to scientists set by PLOS. We can probably assume even without this experiment that dopamine manipulations affecting cognition (like attention, motivation, and learning) will impact ToM tasks. So, the paper's significance might hinge on uncovering the mechanisms behind dopamine's impact, which the authors attempt to demonstrate with limited success in the secondary analyses. Other reviewers may disagree. Some might believe that it is important to confirm that there is an effect, even though it is likely there would be. 

The secondary conclusion is that 'dopamine modulates inference from mental- and non-mental state animations via independent mechanisms' based on two lines of work. It seems like the authors are diving into some exploratory territory, which can be exciting but also raises some conventional but considerable questions. For instance, why would our motor responses or emotion recognition be more relevant to understanding intentions in the 'mental state' category compared to the 'non-mental state' one? It's not immediately clear from a theoretical standpoint. Moreover, the Bayesian approach also makes it difficult to control for multiple comparisons. 

The first claim is that haloperidol diminishes the specificity of motor similarity use for the 'mental state' category attribution accuracy. This diminishing effect is considered by the authors to be a specific reduction of the use of 'motor similarity' for the 'mental state' condition (a contrast between PLA and HAL). Was that comparison done? I could not find it. 

For consideration, I tested the reported 3-way interaction with a (frequentist) linear mixed model equivalent of the analysis, and it is not significant (p = 0.076, regardless of method). Frustratingly, there is no way to provide a regression table in a review. It becomes less significant if the mental state is added as a random slope or if the video is not a random grouping factor (this is not the best fitting model presumably, but it just suggests the finding may not be the most reliable). So, I consider this effect with some skepticism, but there is probably a trend effect here. I do not wish to open a Bayesian vs frequentist debate; I just think the authors and editors consider that the two statistical approaches disagree, and what should a reader do with that information when the study was not preregistered. 

Now, onto the graphs. I noticed something interesting in Figure 2—the jerk difference in the 'mental state' condition seems to hit a ceiling around z = 5.3, while the 'non-mental state' condition doesn't show the same pattern. What's going on there? Does this difference in distribution affect the analysis? 

On lines 341-347 (Model 3, placebo jerk), the authors suggest that due to the absence of an interaction [between jerk difference (motor similarity), mental state, and drug], an effect of placebo jerk difference on mental state animations can be inferred for both placebo (PLA) and haloperidol (HAL) trials. Specifically, they suggest that under haloperidol administration, participants' attribution accuracy is influenced by their movements in the placebo condition rather than movements produced under the drug condition. The authors do need to be careful here. The absence of a difference between HAL and PLA conditions does not signify their equivalence. As proponents of Bayesian analysis, the authors could ascertain whether sufficient evidence exists for the absence of a difference between PLA and HAL conditions to make that inference. Additionally, an investigation solely within the HAL condition to determine if placebo-jerk difference serves as a significantly superior predictor compared to HAL-jerk difference is warranted. As it stands, Figure 2C looks very similar to Figure 2B. By completing those analyses the authors could solidify their conclusions: that HAL-treated participants rely more on placebo-based motor inferences than HAL-based ones, and similar to placebo. 

Including HAL jerk differences in the online data could really help when it comes to reviewing the information more closely.

Another point of the authors supporting the claim about two different paths is the connection to emotion regulation. On line 383, the authors mention, "If we see specific changes in how drugs affect recognizing emotions and understanding mental states, but not for non-mental state animations, it could show how dopamine affects our social understanding." However, I respectfully disagree with this because both types of videos likely involve our social understanding, as I noted in my earlier review.

When we look at the secondary analyses together, it seems that haloperidol mainly affects the 'mental state' category and related measures. This includes both individual differences in how we perceive ourselves and others, as well as recognizing emotions. Initially, I thought the abstract suggested they found separate effects for each category, but it turns out they both hinge on distinguishing between 'mental state' and 'non-mental state'. Could it be that haloperidol affects motivation or effort rather than independent mechanisms? Especially since the 'mental state' category is known to be more challenging and might require more focus?

My more fundamental issue with the secondary conclusions is that both categories require understanding mental states, and the line between them isn't clear. The authors did a great job summarizing previous research on these conditions. While I can see that they might involve subtly different levels of mentalizing, labeling them as 'mental state' and 'non-mental state' will cause confusion. Typically, the literature compares tasks involving mentalizing to those that don't, rather than two tasks with slight differences in mentalizing. Also, given that both categories involve inferring mental states, it might be misleading to suggest that 'non-mental state' tasks don't involve understanding others' minds. This is what is implied by the authors on line 383 (that only one category involves social cognition). 

My main concern is that there doesn't seem to be a clear theoretical difference between the categories. Without well-defined criteria, it's hard to claim separate pathways for different categories, as the authors suggest. Instead, the differences in effects might reflect varying levels of effort, attention, or other factors. Upon reflection, it's worth considering that these animation categories differ in various ways, like how the triangles interact, move, and demand attention. These differences might explain why one task is easier than another and its relation to other tasks.

One solution could be to reconsider the labels used or even drop this comparison altogether. It might be best to do so if it's unclear why these categories differ in terms of participants' cognition. That said, I appreciate that the primary finding may not be compelling enough on its own, so the authors may want to try to flesh out what is really different between these video categories, and go from there -- as long as they can justify the results in terms of exploratory analysis (without correction for multiple comparisons, if appropriate etc). 

Lastly, regarding the effectiveness of blinding, was there any way to assess this?

On line 41, it might be worth noting that there's no 'published' study on this topic.

Reviewer #3: Thankyou to the authors for responding to all of my comments. I am happy that they have all been addressed. 

Reviewer #4: 

I have been asked to review this revision with a special focus on the Bayesian analyses. In brief, I thought the analyses were entirely appropriate. You might consider orientating the reader to the nature of you

---

## [Editor Report · Decision Letter 3]

17 Apr 2024

Dear Dr Schuster,

Thank you for your patience while we considered your revised manuscript "Dopamine challenge reduces mental state attribution accuracy" for publication as a Research Article at PLOS Biology. This revised version of your manuscript has been evaluated by the PLOS Biology editors and the Academic Editor.

Based on on our Academic Editor's assessment of your revision, we are likely to accept this manuscript for publication, provided you satisfactorily address the following data and other policy-related requests.

* We would like to suggest a different title to improve clarity: "Disruption of dopamine system function impairs human ability to understand the mental state of other people"

DATA POLICY:

Regardless of the method selected, please ensure that you provide the individual numerical values that underlie the summary data displayed in the following figure panels as they are essential for readers to assess your analysis and to reproduce it: Figure 1 and the supplementary figure.

CODE POLICY

Per journal policy, if you have generated any custom code during the curse of this investigation, please make it available without restrictions upon publication. Please ensure that the code is sufficiently well documented and reusable, and that your Data Statement in the Editorial Manager submission system accurately describes where your code can be found. 

Could you please also add a readme file to the OSF repository? It is currently a bit challenging to understand the files and data that you provide there.

We expect to receive your revised manuscript within two weeks. 

*Published Peer Review History*

*Press*

Sincerely,

Christian

Christian Schnell, PhD

Senior Editor

cschnell@plos.org

PLOS Biology

---

## [Editor Report · Decision Letter 4]

1 May 2024

Dear Dr Schuster,

Thank you for the submission of your revised Research Article "Disruption of dopamine system function impairs the human ability to understand the mental states of other people" for publication in PLOS Biology. On behalf of my colleagues and the Academic Editor, Raphael Kaplan, I am pleased to say that we can in principle accept your manuscript for publication, provided you address any remaining formatting and reporting issues. These will be detailed in an email you should receive within 2-3 business days from our colleagues in the journal operations team; no action is required from you until then. Please note that we will not be able to formally accept your manuscript and schedule it for publication until you have completed any requested changes.

PRESS

Sincerely, 

Christian

Christian Schnell, PhD

Senior Editor

PLOS Biology

cschnell@plos.org